# Development and Psychometric Validation of the Breast Cancer Stigma Assessment Scale for Women with Breast Cancer and Its Survivors

**DOI:** 10.3390/healthcare12040420

**Published:** 2024-02-06

**Authors:** Judit Cenit-García, Carolina Buendia-Gilabert, Carmen Contreras-Molina, Daniel Puente-Fernández, Rafael Fernández-Castillo, María Paz García-Caro

**Affiliations:** 1Virgen de las Nieves University Hospital, 18014 Granada, Spain; juditcenit@correo.ugr.es (J.C.-G.); carmen.contreras.sspa@juntadeandalucia.es (C.C.-M.); 2ibs.GRANADA—Biosanitary Research Institute, 18012 Granada, Spain; 3Medical Oncology Unit, Vall d’Hebron University Hospital, 08035 Barcelona, Spain; carolina.buendia@vallhebron.cat; 4Department of Nursing, Faculty of Health Sciences, University of Granada, Avda. de la Ilustración, 60, 18016 Granada, Spain; rafaelfernandez@ugr.es (R.F.-C.); mpazgc@ugr.es (M.P.G.-C.); 5CIMCYC—Mind, Brain and Behavior Research Center, University of Granada, 18071 Granada, Spain

**Keywords:** breast neoplasm, cancer, social stigma, patients, cancer survivors, validation study

## Abstract

Background: The increase in breast cancer cases and breast cancer survival makes it advisable to quantify the impact of the health-related stigma of this disease. Purpose/Objectives: To develop and validate a breast cancer stigma scale in Spanish. Methods: Women diagnosed with, or survivors of, breast cancer were included. The development of the Breast Cancer Stigma Assessment Scale (BCSAS) involved both a literature review and personal interviews. Content validity was assessed using a Delphi study and a pilot test; construct validity was evaluated using an exploratory factor analysis; and convergent validity was assessed using six scales. Cronbach’s α internal consistency and test-retest reliability were used to determine the reliability of the scales. Results: 231 women responded to the 28-item scale. The BCSAS showed good reliability, with α = 0.897. Seven factors emerged: concealment (α = 0.765), disturbance (α = 0.772), internalized stigma (α = 0.750), aesthetics (α = 0.779), course (α = 0.599), danger (α = 0.502), and origin (α = 0.350). The test-retest reliability was 0.830 (*p* < 0.001). Significant correlation was observed with event centrality (r = 0.701), anxiety–depression (r = 0.668), shame (r = 0.645), guilt (r = 0.524), and quality of life (r = −0.545). Conclusions: The BCSAS is a reliable and valid measure of stigma in women with breast cancer and its survivors. It could be useful for detecting stigma risk and establishing psychotherapeutic and care priorities.

## 1. Introduction

Breast cancer, particularly female breast cancer (FBC), is the most commonly diagnosed cancer in the world [1], and has the highest incidence, as well as the greatest number of years lived with disability rate in Europe [2], with an estimated 35,000 new cases predicted in Spain by 2023 [3]. Additionally, the number of cases is expected to increase. But survival has also increased to 86% worldwide and to 90.6% in Spain [3]; this implies that more and more women will suffer from breast cancer and survive it, thus facing its impact, which extends to all spheres of life.

Cancer remains a profoundly feared disease, and it is associated with social representations [4] of death, disability, physical disfigurement or deformity, suffering, and pain [5,6,7]. These social representations, as well as experienced or anticipated realities (breast deformities; scarring; alopecia; lymphedema; and work, social, economic, family, and relationship problems), result in the emergence of stigma [8,9,10].

Stigma can be defined as a specific contextual relationship in which a particular attribute is associated with a negative evaluation that can lead to negative treatment or discrimination, as well as self-fulfilling prophecies, the activation of stereotypes, and identity threat [11]. Stigma is internalized or perceived when negative evaluations are self-inflicted and internalized, and it can lead to shame, guilt, and/or the fear of being discriminated against. Social or public stigma refers to an induced negative evaluation, as opposed to real discrimination [12].

Health-related stigma (HRS) is a social, experienced, or anticipated process characterized by exclusion, rejection, blame, or devaluation that results from a negative evaluation, with a permanent alteration in identity that is caused by a health problem. This evaluation may be medically unjustified and may worsen the patient’s health status [13]. HRS has been studied in HIV, mental illness, epilepsy, physical disability, and lung cancer patients [13,14]. According to the literature, it acts as an enhancer of the burden of illness for patients and their families and is associated with a delayed presentation for care, the premature termination of treatment, limitations in access to services and resources, and amplified psychological, physical, and social morbidity [8,11,13], serving as a predictor of psychiatric and psychological comorbidities.

According to Fujisawa and Hagiwara’s model [15], cancer is a stigmatizing condition when the following elements of stigma described by Jones [16] are present: concealability, course, disturbance, steric quality, origin, and danger.

Models of stigmas [15,17,18,19,20] explain their influence on the behavior of the stigmatized and those with whom they interact, as well as the threat or damage to the individual’s social identity (Figure 1). Stigma causes increased stress, emotional distress, poor coping, and decreased self-efficacy [6,7,19,21]. In the serial mediation model of stigma in breast cancer (Zamanian, 2022), the importance of the relationship between public stigma, internalized stigma, body image, and psychological distress was confirmed [20].

Previous studies have placed the prevalence of perceived cancer-related stigma in general at between 5 and 90% [22]; for breast cancer, 85.4% of survivors reported moderate–high levels of stigma [23].

On the other hand, there is sufficient evidence supporting associations between breast cancer and fatigue, distress, depression, and anxiety months or even years after the diagnosis [24]. The overall prevalence of depression in women with breast cancer in 70 countries has been estimated at 32.2%, with similar figures for anxiety [25]. In Spain, the prevalence of anxiety has been placed at 48.6%, and at 15% for depression [26,27]. This prevalence varies according to the stage of breast cancer, the time of treatment, recurrence, and other factors [25], including the social and cultural factors of stigma, with a higher incidence in patients suffering from anxiety and depression in general [28,29,30] and in women with breast cancer in particular [24,25,31,32]. In addition, specific stigma interventions have had an effect on psychological outcomes [33], suggesting that they may be secondary to the experience of stigma [13,21]. The associated consequences have been described in terms of increased disability, a poorer quality of life, and the decreased effectiveness of general resilience (sense of coherence, social support, and coping skills) [34,35].

Likewise, some reviews and meta-analyses in recent years have corroborated the strong relationship of stigma in women with breast cancer with negative body image, anxiety, resignation, depression, guilt, ambivalence about emotional expression, and social restriction, as well as with delayed help-seeking behavior [36,37]. Studies reveal that there are numerous sociodemographic variables, including the disease itself and its treatments, along with psychosocial variables that are related to stigma, causing suffering and related both to the conditions inherent to the disease experience and to the recovery of these women. Health professionals should pay attention to the assessment of stigma in order to respond to the needs of patients and survivors, as well as to develop better strategies for breast cancer health promotion and prevention [36,37,38,39,40,41,42,43,44].

Over the last two decades, instruments have been developed to quantify the impact of health-related stigma, including the Internalized Stigma of Mental Illness (ISMI) scale [45] and the HIV Stigma Scale [46,47]. Based on the latter, the Cataldo Lung Cancer Stigma Scale (CLCSS) was developed in 2011 [14], and recently, instruments have been designed to specifically measure stigma in Iranian and Chinese women with breast cancer (Dewan et al. 2020 and Bu et al. 2022) [34,35]. However, these tools are designed for patients in the active phase. During this phase, the patient is focused on the consequences of cosmetic changes. These scales do not include other relevant items for the assessment of stigma or its impact on survivors. Moreover, we have not found any instrument developed in the Spanish population that assesses breast cancer stigma, nor do currently used instruments include all relevant aspects of stigma, apart from the aesthetic items, applicable to the assessment of stigma or its impact on survivors.

The comorbidities associated with stigma justify the need to assess its role as a mediator or potential cause of decreased resilience and quality of life, the amplification of morbidity, and worsening prognosis. Stigma research will be useful in promoting and evaluating specific interventions that respond to the needs of women with breast cancer and its survivors.

The aim of this study is to develop and validate a scale sensitive to the stigma experiences of Spanish women with breast cancer and its survivors to explore its impact, incidence, duration, evolution, and other related factors.

## 2. Materials and Methods

### 2.1. Design

A descriptive, cross-sectional study was carried out for the development and validation of an instrument for stigma evaluation. This study was divided into three phases: (1) item development, (2) scale development, and (3) scale evaluation, following the recommendations for developing and validating scales [48,49,50].

### 2.2. Phase 1: Item Development

The development of the scale items was based on the health-related stigma (HRS) theoretical model created by Weiss [51] and was developed for cancer stigma evaluation in the previously described conceptual framework created by Fujisawa and Hagiwara [15]. 

#### 2.2.1. Literature Review

First, an in-depth review of the literature published up to that time (December 2019) in *PubMed*, *Web of Science*, *Embase*, and *CINAHL* was performed. Search strategies combining breast cancer, stigma, social stigma, and variations were employed. The results showed that there was no scale to assess stigma in breast cancer, with the HIV Stigma Scale [46] (validated in Spanish [47]) and the Cataldo Lung Cancer Stigma Scale [14] being the closest stigma scales currently available.

In parallel, a qualitative study was conducted with the participation of 15 women with breast cancer, or survivors of the disease, over 18 years of age, in a disease-free period, during remission, or during a follow-up at the gynecologic oncology unit of the Virgen de las Nieves Hospital in Granada, selected through convenience sampling, in order to explore stigma in breast cancer and to identify related factors. The sample size was reached at the saturation of the data. The interviews were conducted in a private office using an individual, semi-structured interview. The interview script was designed so that the discursive elements surrounding stigma experiences, as well as those related to stigma responses, could be addressed in a comprehensive manner: (1) social imaginary; (2) reactions, projections, and attributions; (3) social experience; and (4) perception of stigma. The duration of the interviews ranged from 60 to 90 min. They were audio-recorded and transcribed verbatim. The interviews were conducted by a researcher trained and experienced in conducting in-depth interviews. Data collection was terminated when data saturation was considered to have been reached [52]. Braun and Clarke’s six-phase framework [53] was used to analyze the data. Rigor was ensured using the criteria of credibility, transferability, reliability, and confirmability [54]. The obtained results were in accordance with the elements of stigma considered in the theoretical framework and were concretized as follows: symbology, metaphors and negative social representations, control of information, guilt in the search of the origin and meaning of the disease, alterations to social and family life, deterioration of image and self-concept, perceived stigma or discrimination, permanent alteration of identity, and negative emotional and behavioral responses.

Subsequently, the first version of the Breast Cancer Stigma Scale was developed.

#### 2.2.2. Item Composition

The procedure for the selection of the items for the first version of the scale was based on a selection of items from the Cataldo Lung Cancer Stigma Scale (CLCSS) [14], after a translation and cultural adaptation of the scale to Spanish and to breast cancer, and from the HIV Stigma Scale, after an adaptation of the scale to breast cancer (validated in Spanish) [46,47,55], along with statements obtained from qualitative interviews.

Eleven items were selected from the CLCSS (only one of which was not shared with the HIV scale), and the items specifically related to smoking-related stigma were discarded, since breast cancer stigma is not specifically based on the belief that the disease is smoking-related. However, four other items from the HIV scale—discarded by Cataldo in his adaptation of the HIV Stigma Scale for his CLSS—were added, whose adaptation for breast cancer was in agreement with the results of the qualitative research and with the theoretical framework. Twenty-one items were selected from the interview results. Some items were considered to reflect the same aspect of stigma, so they were merged to improve the operability of the scale. Likewise, the verb tense and the form of expression were adapted to be independent of the time of the disease, whether in the active phase or in a period of remission.

A total of 36 items (4 based on the HIV stigma scale, 11 based on the CLSS, and 21 based on the interviews) comprised the first version of the scale.

#### 2.2.3. Delphi Study

The first version of the Breast Cancer Stigma Assessment Scale (BCSAS) underwent two rounds of a Delphi study [56]. Fifteen specialists from the fields of anthropology, sociology, psychology, oncology, and nursing, with a mean work experience of 19.8 years, were contacted via email. These experts evaluated an online questionnaire using a Likert scale of 1–4 (where 1 was disagree, and 4 was completely agree) to assess the adequacy of the items in relation to the concept of stigma in terms of relevance, pertinence, clarity, and completeness, and suggestions were encouraged on their part. For the evaluation of the level of agreement among experts, a score of 3 or 4 obtained for each item was calculated. A minimum consensus level of 80% was established.

Items rejected by more than one of the fifteen reviewers were discarded or rewritten. Of the 36 initial items, 28 were remained, with a 93.3% degree of consensus in the second round. 

### 2.3. Phase 2: Scale Development

#### Pilot Testing

With the second version of the scale, including 28 items, a pilot test was carried out with the participation of 62 women, over 18 years of age, with breast cancer in the active stage or in remission, who attended a consultation in the gynecologic oncology department or who belonged to previously contacted associations for women with breast cancer. They answered the online questionnaire, in which they were also asked for their opinions regarding the questionnaire.

The pilot sample included 62 participants, resulting in α = 0.883 (95% CI = 0.836–0.921) with a Kaiser–Meyer–Olkin (KMO) value of 0.746 and a Bartlett’s sphericity test value of χ^2^ = 758.981 (*p* = 0.001). This indicated that the BCSAS exhibited good internal consistency, and the exploratory factor analysis was relevant.

After the pilot sample, the final scale was maintained using the 28 items of the second version, and the second phase of the validation of the BCSAS was carried out in the Spanish population.

### 2.4. Phase 3: Scale Evaluation

#### 2.4.1. Sample and Setting

Women were recruited intentionally and consecutively among those who had been diagnosed with breast cancer between 1 January 2006 and 31 October 2022; were of Spanish nationality; were undergoing treatment, follow-up, revision, or remission of their disease at the Complejo Hospitalario Virgen de las Nieves of Granada (Spain); were recruited through associations of women with breast cancer; possessed the cognitive capacity to complete the questionnaire; and who had provided written informed consent.

#### 2.4.2. Data Collection

The first author contacted the participating centers and requested their collaboration in the recruitment of patients between June 2021 and March 2022. In each of the centers, the participation of an oncologist or oncology consultation nurse, who recruited women that met the inclusion criteria and agreed to participate in the study, was requested. The collaboration of associations of women with breast cancer throughout Spain was requested by means of an e-mail including information, documentation, instructions, participation criteria, and contact details for the investigators.

After explaining the objective of the study and what their participation consisted of, the participants were provided with a web link to access the questionnaire to be completed. The completion of the entire questionnaire required between 35 and 45 min. A total of 70 patients repeated the BCSAS questionnaire 3–5 weeks after the first completion to evaluate its test-retest reliability.

#### 2.4.3. Tools

An online form was developed using the LimeSurvey software application (V 5.6.31+230718) LimeSurvey GmbH [57], which included sociodemographic data (age, marital status, economic level, educational level, and employment status) and clinical data (year of diagnosis, stage, and course of disease), in addition to the following instruments:

The Breast Cancer Stigma Assessment Scale (BCSAS): this scale includes 28 items, requiring rating the statements regarding stigma experiences using a 5-point Likert scale (1 = strongly disagree, and 5 = strongly agree). It was scored between a minimum of 28 and a maximum of 140; the higher the score, the greater the degree of stigma (see Appendix A). 

The Hospital Anxiety and Depression Scale (HADS) [58]: this scale was chosen due to its proven usefulness in the oncology population [59] and the availability of a validated version in Spanish for use with this population [60]. It is a 14-item scale with two subscales relating to anxiety and depression. The total HADS score ranges from 0 to 42. The odd-numbered items make up the anxiety subscale, and its response scale is scored from 3 to 0. The even-numbered items make up the depression subscale and are scored from 0 to 3. The total score in each subscale is obtained by adding those of the corresponding items, with each item ranging from 0 to 21. In both cases, the higher the score, the higher the level of anxiety or depression. This instrument has been validated in the Spanish oncology population, with a Cronbach’s alpha reliability of 0.85 for the anxiety subscale and 0.87 for the depression subscale [57].

Quality of Life Scale (SF-12) [61]: this is an instrument that assesses the health-related quality of life. It consists of twelve items integrated into eight dimensions grouped into two summative components: physical health (score range: 6–20) and mental health (score range: 6–27), yielding a total CDVRS score (score range: 12–47). Higher scores indicate a better CDVRS [62]. It is demonstrably useful for oncology patients [63,64]; the reliability indices obtained for oncology patients were as follows: physical health, α = 0.75; mental health α = 0.78; and CDVRS α = 0.83 [61]. For a sample of surviving breast cancer patients, the physical health α = 0.74, the mental Sa-health α = 0.77, and the CDVRS α = 0.88 [64].

Centrality of Event Scale (CES) [65]: this scale is used to assess the experience of an event as a turning point in life, when it is part of the person’s identity and personality and when the event becomes a reference for attributing meaning to other experiences. It is a unifactorial scale composed of 20 items, with a response range between 1 (completely disagree) and 5 (completely agree). The higher the score, the greater the centrality of the event, with scores ranging from 20 to 100. The original and Spanish validation [66] presented an internal consistency Cronbach’s alpha of 0.94, 0.92, and 0.94, respectively (with a double sample in the Spanish validation). The correlations show a positive and significant relationship between the event centrality and the state anxiety, trait anxiety, depression, and post-traumatic stress symptomatology [66].

Experience of Shame Scale (ESS) [67]: this is an instrument that assesses shame, as one of the emotions associated with self-awareness, organized into three areas: characterological shame, behavioral shame, and bodily shame. It consists of 25 items, scored from 0 to 4, yielding total scores in the range of 25–100. The Spanish adaptation of this scale showed a Cronbach’s alpha of 0.91 [68].

Guilt Scale (SC-35): this is an instrument that measures the concept of guilt, understood as a process of appraisal (cognitive and affective) of those behaviors that are not in accordance with a certain scale of values. It specifically evaluates the ease of or propensity to feel guilt. It consists of 35 items scored on a Likert scale of 1 to 5 points, from 1 (totally false) to 5 (totally true). The scores are: “low guilt” (0–40 points), “normal guilt” (41–100 points), “excessive tendency to blame” (101–120 points), and “pathological guilt” (>121 points). This scale has proven to be a reliable instrument (Cronbach’s alpha = 0.88) [69].

#### 2.4.4. Data Analysis

An analysis of the sociodemographic and clinical data was performed by calculating the frequencies and percentages for the qualitative variables, as well as the means and standard deviations for the quantitative variables.

For the exploratory factor analysis, the KMO test and Bartlett’s test of sphericity were performed. Next, a principal component analysis, with varimax rotation, was performed.

Cronbach’s alpha, with confidence intervals, was calculated for the total scale and for each of the factors resulting from the exploratory factor analysis. For intraobserver reliability, the intraclass correlation coefficient (ICC) was calculated. An ICC < 0.50 indicated poor to fair agreement, an ICC = 0.5–0.75 indicated good agreement, and an ICC > 0.90 indicated excellent agreement [67].

For convergent validity, the normality of the variables was tested using the Kolmogorov–Smirnov test. Due to the non-normality of the variables, Spearman’s rho correlation coefficient, with 95% confidence intervals, was determined. An r < 0.39 indicated a weak correlation, an r = 0.40–0.69 indicated a moderate correlation, and an r > 0.7 indicated a strong correlation [68].

The feasibility was calculated by measuring the mean response time in minutes for a selection of 20% of the sample.

The data analysis was performed using IBM SPSS v28 statistical software [70].

### 2.5. Ethical Consideration

The study was conducted according to the guidelines of the Declaration of Helsinki. The data were anonymized prior to analysis, in line with the requirements of Spanish Data Protection Act 2/2018, including all data relevant to the study, removing all traces of personal information. The study was approved by the Research Ethics Committee for the Andalusian Public Health System in Granada (verification code: 3abcc9372346671adb0d935d9d3cc2af23810429). All participants provided informed consent.

## 3. Results

### 3.1. Characteristics of the Study Participants

A total of 231 women responded to the questionnaire. The mean age was 51.15 years, with an SD = 10.369. They were mainly married (61.5%) and working (46.3%), with a self-perceived average economic level of 81.4% and a vocational (29.9%) or university (52.9%) education. Most of the patients had stage II cancer (31.6%), and the majority were cured or free of disease (48.5%) (Table 1). Regarding feasibility, the mean response time assessed in 68 participants was 7.10 min (SD = 5.39).

### 3.2. Construct Validity

The value of the KMO was 0.884, and Bartlett’s sphericity test value was χ^2^ = 2211.935 (*p* = 0.001), indicating that the sample was appropriate for factor analysis. The BCSAS scale showed good reliability, with α = 0.897 (95% CI = 0.878–0.915), and seven factors: Factor 1: “Concealability” = 0.702 (95% CI = 0.638–0.754); Factor 2: “Discrimination” = 0.772 (95% CI = 0.720–0.817); Factor 3: “Altered self-image and self-concept” = 0.750 (95% CI = 0.695–0.797); Factor 4: “Family Disruption” = 0.779 (95% CI = 0.725–0.825); Factor 5: “Social attributions” = 0.599 (95% CI = 0.508–0.677); Factor 6: “Prejudices” = 0.596 (95% CI = 0.497–0.679); Factor 7: “Origin” = 0.587 (95% CI = 0.465–0.682) (Table 2).

### 3.3. Reliability

For each item, statistically significant test-retest reliability was obtained. Score correlation ranged from 0.438 to 0.907 for test-retest reliability (Table 3). A test-retest reliability ICC = 0.830 (95% CI: 0.719–0.837, *p* = 0.001) was observed for the total score scale.

### 3.4. Convergent Validity

A positive correlation was observed between the BCSAS and its different dimensions with the HADS, r = 0.668 (95% CI = 0.587–0.736); CES, r = 0.701 (95% CI = 0.625–0.764); ESS, r = 0.645 (95% CI = 0.555–0.720); and SC-35, r = 0.524 (95% CI = 0.415–0.617). A negative correlation was observed between BCSAS and SF12; r = −0.545 (95% CI = −0.634–−0.441) (Table 4).

## 4. Discussion

This study aimed to develop and validate the Breast Cancer Stigma Assessment Scale (BCSAS). The developed scale obtained a high degree of consensus for content validity, and the final version presented an adequate factor structure and internal consistency. In addition, it showed good results for internal reliability, as well as test-retest reliability. It correlated positively with the HADS, ECE-SF, ESS, and SC-35 scales, and negatively with the SF12. These results indicate that the BCSAS is a valid instrument for assessing stigma in Spanish breast cancer patients and survivors, with possible generalizability in other countries with similar language and cultural contexts.

The BCSAS scale consists of 28 items and comprises seven factors: concealability, discrimination, image and self-concept disruption, family disruption, social attributions, prejudice, and origin. These factors are in line with Fujisawa and Hagiwara’s conceptual model of cancer stigma [15], but they present specific aspects applicable to breast cancer in Spanish women. Thus, for example, the disruption factor in our population was centered on the family and couple environment, or the course factor, which took the form of social attributions and prejudice, while the danger factor was less significant.

To our knowledge, two recently published scales have been developed to specifically assess breast cancer stigma. The first is the Breast Cancer Stigma Scale for Arab patients (BCSS-A), published in 2020 [34]. It is a questionnaire, without factorial structure, composed of 12 items. This scale was developed using a sample of 59 Muslim women undergoing active cancer treatment, almost all of whom were married. The BCSS-A showed correlation only with depressive symptomatology. This scale has important methodological and conceptual limitations that prevent us from generalizing its findings and, therefore, from comparing its results with those from our study.

The second scale, the Breast Cancer Stigma Scale (BCSS), published in 2022 [35], was developed and validated in Chinese patients diagnosed with breast cancer, who were undergoing treatment, using a sample of 200 women. The authors developed a scale of 15 items and four factors (impaired self-image, social isolation, discrimination, and internalized stigma) based on the conceptual model of perceived stigma related to lung cancer, developed by Cataldo et al. [14], and the scale was correlated with the Fife and Wright Social Impact Scale [71] as a criterion for validity.

With respect to the BCSS validated in Chinese patients, the BCSAS developed and validated in this study presents differences at various levels, which can be considered strengths. First, the women participating in the BCSAS were female patients at different points of the disease, along with female survivors (cured or disease-free); thus, the perspective of stigma was broader, allowing the evolution of stigma over time to be evaluated, including the impact of sequelae, limitations, and other consequences of a permanent nature.

Secondly, the BCSAS is organized into seven factors with 28 items, based on the general HRS framework [13,51] and on the specific framework of cancer stigma [15], while the BCSS includes four factors with 15 items, based on Cataldo’s model of perceived stigma [14], which is based in turn on Berger’s HIV model [46]. A detailed analysis of the BCSS items and the included factors revealed that this scale, designed to measure the perceived stigma, is more oriented to internalized stigma than to perceptions of how women with breast cancer are treated; in addition, the items related to the deterioration of self-image (the theoretical equivalent of aesthetics in the HRS) are focused on breast deformities after surgery, while other concerns of great impact to body image, such as alopecia secondary to chemotherapy treatment or changes in body weight, are not reflected. For some women, losing their hair was found to be even more distressing than losing their breasts [10]. The social isolation factor assesses intimate contact, but not social contact; it only includes one item regarding concealability, i.e., the information and visibility of their disease. Moreover, other elements of cancer stigma [15] are not evaluated, such as job discrimination, the possible guilt associated with the belief that cancer is a self-inflicted disease (origin factor), and family and social disruptions that hinder or block interactions and communication, in addition to intimate contact (disruption factor).

Finally, the negative correlation with the quality of life assessed by the SF-12 scale [64] also reinforces the convergent validity and is in agreement with similar studies that associate a higher degree of stigma with a lower quality of life; a meta-analysis [36] revealed a negative correlation in the same way as that observed for the BCSAS scale. Other studies have confirmed this association [72,73,74].

With respect to the consistency and validity of our scale, BCSAS has an excellent internal consistency, measured with a Cronbach’s alpha of 0.897. This result is superior to those shown by other instruments that evaluate the same construct, such as the 15-item Chinese BCSS scale (alpha of 0.86) [35]. Response stability is another strength of the BCSAS scale. Good test-retest reliability of the total score was observed, with an intraclass correlation index of 0.830 [75].

Regarding the convergent validity of the BCSAS scale when compared with instruments that evaluate similar constructs, a statistically positive correlation was observed (r = 0.701) with the degree of centrality of the traumatic event (CES) [66]. This is due to the fact that the subjective experience of a given event is fundamental for it to be considered traumatic. This correlation corroborates the relationship between the experience and the interpretation of breast cancer as a traumatic event leading to an alteration of the implicit identity, resulting in stigma. This is in line with the results of research that identified high correlations between post-traumatic stress disorder and event centrality in women with breast cancer [65]. Wong also negatively associated self-stigma in breast cancer with posttraumatic growth and its mediating (protective) role on the effects of self-stigma on quality of life [72].

The anxiety and depression construct was also assessed using the (HADS) scale [59]. A statistically significant correlation (r = 0.668) was observed between the BCSAS and the HADS scales [76]. This observed correlation could be due to the fact that stigma is a possible cause and amplifier of anxiety and depression through increased distress, resulting in psychological and psychiatric comorbidities in women with breast cancer, which is widely described by the literature [24,26,32,36].

Another construct evaluated concerned shame (ESS) [67,68] and guilt (SC-35) [69], which are considered central dimensions of stigma [12,13,15]. A correlation of r = 0.645 (95% CI = 0.555–0.720) was observed for shame and r = 0.524 (95% CI = 0.415–0.617) for guilt, data which are in agreement with the correlations found in Tang’s meta-analysis [36].

Finally, the negative correlation with the quality of life assessed by the SF-12 scale [61] also reinforces the convergent validity and is in agreement with similar studies that associate a higher degree of stigma with a lower quality of life; a meta-analysis [36] revealed a negative correlation similar to that observed for the BCSAS scale. Other studies confirm this association [20,21,38].

The validation of the Breast Cancer Stigma Assessment Scale (BCSAS) confirms it as a tool to measure and evaluate the impact of stigma as a possible cause of decreased resilience, self-esteem, self-efficacy, and quality of life; the amplification of psychological and social morbidity; and the worsening of prognosis in Spanish women with breast cancer and survivors of the disease, as well as in those reflecting similar linguistic and cultural contexts. The availability of this instrument could facilitate the incorporation of stigma into the health policy agenda, serving to promote and evaluate specific interventions for stigma in breast cancer. Health professionals can employ this instrument for the provision of services and the improvement of the health of these women.

As limitations of this work, the fact that the sample of women who participated was intentional and collected in hospital centers, as well as from associations of women with cancer, should be considered. This may introduce some type of selection bias. One of the characteristics of stigma is the refusal to participate in groups or activities with other people with cancer, so it is possible that some of the women who did not want to participate in this research were not part of the sample for this reason.

As for future lines of research, it would be advisable to implement diagnostic and intervention programs regarding stigma and its subsequent evaluation, which could result in improving the quality of care for women with breast cancer.

## 5. Conclusions

The BCSAS is an appropriate tool for the assessment of stigma in women with breast cancer and its survivors, and it provides useful information regarding the impact, evolution, and dimensions of stigma in such a way that it fulfills the dual function of quantifying and characterizing stigma, guiding the potential areas of focus in health care.

It is imperative that health professionals incorporate stigma assessment into their interventions. The associated comorbidities and the additional suffering involved for women with breast cancer and its survivors are evidence of this. This consideration will enable individualized responses and the development of health policies for breast cancer health promotion and stigma prevention.

## Figures and Tables

**Figure 1 healthcare-12-00420-f001:**
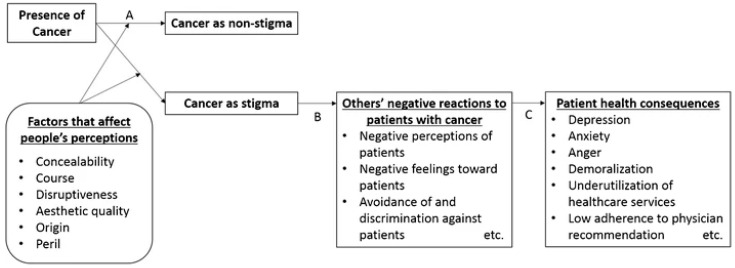
Fujisawa and Hagiwara’s conceptual model of cancer stigma [15]. A: The presence of cancer in itself is not always considered as a stigma, particularly when it does not convey a deviation from the norm and/or is not associated with undesirable qualities. B: The presence of cancer, once labeled as a stigma, can induce negative reactions among the general population toward the patients. C: Negative reactions from others can impact patient health status as well as health-related behaviors.

**Table 1 healthcare-12-00420-t001:** Sample characteristics.

Variable	Group	n	*p* (%)
Age	X = 51.69 (SD = 10.6)		
Marital status	Single	20	8.7
	Married	142	61.5
	Separated/divorced	24	10.4
	Widowed	8	3.5
	Cohabiting with partner	37	16.0
Self-perceived economic level	Very High	1	0.4
	High	24	10.4
	Medium	188	81.4
Level of education	Low	18	7.8
	No education	2	0.9
	Primary school	20	8.7
	High school	18	7.8
	Middle/higher vocational training	69	29.9
	University	122	52.8
Employment status	Active	107	46.3
	Low Labor	42	18.2
	Unemployed	17	7.4
	Retired	49	21.2
	Other	14	6.1
Stage	I	51	22.07
	II	73	31.6
	III	56	24.24
	IV	14	6.06
Course of disease	Curative treatment	76	32.9
	In remission	31	13.4
	Cured/disease-free	112	48.5
	In palliative treatment	2	0.9
	Relapse	10	4.3

**Table 2 healthcare-12-00420-t002:** Factor loadings of the BCSAS.

Item	Factor	Total Alpha If Item Is Deleted
Factor 1. Concealability		
1 I hide or minimize my disease with some people.	0.505	0.893
3. I regret having told some people that I have breast cancer.	0.660	0.895
4. In some situations I am embarrassed to say that I have breast cancer.	0.755	0.894
7. I prefer to avoid certain places since I have breast cancer.	0.443	0.891
12. I make an effort to hide or disguise physical changes resulting from breast cancer.	0.529	0.891
17. If I think that I have cancer in my body, I feel disgusted.	0.578	0.896
27. I do not like or avoid participating in groups or activities where I have to be with other people with cancer.	0.458	0.898
Factor 2. Discrimination.		
5. Since breast cancer, I sometimes feel isolated from the rest of the world.	0.527	0.890
6. My breast cancer has a negative or limiting effect on me in my work.	0.759	0.893
9 I feel uncomfortable with the stares, morbidness, or curiosity of some people.	0.446	0.890
11. I feel that I am not as valid as others because I have breast cancer.	0.743	0.893
Factor 3. Altered self-image/self-concept.		
13. When you have breast cancer, hair loss or physical sequelae are a major concern.	0.710	0.896
19. At times I have found it difficult to say and/or hear the word cancer.	0.604	0.893
20. I often feel afraid or worried because I feel in danger because of cancer.	0.578	0.892
21. I feel I am not the same as I was before breast cancer.	0.463	0.892
22. Having cancer has marked a before and after in my life.	0.702	0.893
Factor 4. Family Disruption.		
23. Having breast cancer harms sexual relations.	0.797	0.891
24. Having breast cancer interferes with family relationships.	0.552	0.893
25. Having breast cancer negatively affects relationships with a partner.	0.825	0.892
Factor 5. Social attributions		
8. I have been bothered by some attitudes or behaviors of people who know about my breast cancer.	0.434	0.890
16. I find it unpleasant that some people feel uncomfortable or avoid me because of breast cancer.	0.642	0.894
18. I don’t like it when people avoid saying or hearing the word cancer.	0.623	0.898
28. I find it hard to face the fact that I may have difficulty or be unable to be a mother in the future because of cancer.	0.593	0.898
Factor 6. Prejudices		
2. I don’t like that some people treat me differently because of breast cancer.	0.722	0.897
26. I worry about how my disease affects the people who care for me.	0.427	0.894
10. I don’t like that some people feel sorry for me.	0.760	0.899
Factor 7. Origin		
14. I believe that my way of being or situations in my life could have caused my breast cancer.	0.757	0.896
15. I think having breast cancer has been a wake-up call that I needed to change some aspects of my life and myself.	0.825	0.900

Total variance explained (%)—58.434.

**Table 3 healthcare-12-00420-t003:** Test-retest reliability of BCSAS.

Factor	Item	M	SD	M′	SD′	ICC	95%	*p*
Factor 1	Item 1	2.94	1.383	2.90	1.291	0.627	0.463–0.749	0.001
	Item 3	2.03	1.113	2.08	1.230	0.438	0.231–0.607	0.001
	Item 4	1.96	1.054	2.04	1.054	0.610	0.442–0.737	0.001
	Item 7	2.85	1.360	2.74	1.30	0.733	0.605–0.824	0.001
	Item 12	2.65	1.313	2.92	1.264	0.635	0.475–0.755	0.001
	Item 17	1.65	0.966	1.61	0.815	0.516	0.325–0.667	0.001
	Item 27	1.97	1.138	2.03	1.210	0.848	0.768–0.802	0.001
	Total Factor 1	16.17	5.26	16.22	5.29	0.867	0.769–0.927	0.001
Factor 2	Item 5	2.67	1.210	2.67	1.311	0.858	0.783–0.878	0.001
	Item 6	3.67	1.353	3.65	1.375	0.837	0.752–0.895	0.001
	Item 9	3.10	1.269	2.93	1.293	0.691	0.548–0.795	0.001
	Item 11	2.94	1.481	2.67	1.511	0.722	0.589–0.816	0.001
	Total Factor 2	12.70	3.98	11.75	4.31	0.857	0.765–0.915	0.001
Factor 3	Item 13	4.21	0.978	4.13	1.020	0.524	0.334–0.673	0.001
	Item 19	3.10	1.484	3.00	1.44	0.848	0.767–0.902	0.001
	Item 20	3.58	1.196	3.60	1.241	0.635	0.474–0.755	0.001
	Item 21	4.13	1.061	4.04	1.106	0.859	0.784–0.909	0.001
	Item 22	4.25	1.031	4.36	0.939	0.702	0.563–0.803	0.001
	Total Factor 3	19.55	3.95	19.09	4.09	0.727	0.570–0.833	0.001
Factor 4	Item 23	3.68	1.287	3.65	1.212	0.784	0.676–0.859	0.001
	Item 24	2.44	1.266	2.40	1.329	0.737	0.610–0.827	0.001
	Item 25	3.31	1.285	3.38	1.305	0.880	0.810–0.923	0.001
	Total Factor 4	9.65	3.25	9.45	3.07	0.816	0.701–0.889	0.001
Factor 5	Item 8	3.35	1.269	3.33	1.364	0.663	0.511–0.775	0.001
	Item 16	3.00	1.343	3.10	1.291	0.593	0.420–0.724	0.001
	Item 18	3.65	1.153	3.49	1.075	0.716	0.582–813	0.001
	Item 28	2.85	1.109	2.88	1.198	0.577	0.400–0.713	0.001
	Total Factor 5	13.19	3.05	12.82	3.12	0.525	0.30–0.696	0.001
Factor 6	Item 2	3.83	1.088	3.88	1.087	0.542	0.353–0.687	0.001
	Item 26	3.93	1.155	3.79	1.138	0.814	0.719–0.880	0.001
	Item 10	4.13	1.061	4.07	1.053	0.749	0.627–0.835	0.001
	Total Factor 6	12.03	2.24	11.98	2.09	0.495	0.261–0.673	0.001
Factor 7	Item 14	2.79	1.299	2.81	1.296	0.686	0.542–0.792	0.001
	Item 15	3.32	1.265	3.33	1.311	0.733	0.604–0.824	0.001
	Total Factor 7	6.15	2.25	6.36	2.09	0.614	0.414–0.757	0.001
	Total Scale	90.21	16.522	87.609	17.009	0.830	0.719–0.837	0.001

ICC: Intraclass correlation coefficient.

**Table 4 healthcare-12-00420-t004:** Convergent validity of the BCSAS.

Scales	HADS		SF12		CES		ESS		SC-35	
	r (95% IC)	*p*	r (95% IC)	*p*	r (95% IC)	*p*	r (95% IC)	*p*	r (95% IC)	*p*
BCSAS	0.668 * (0.587–0.736)	0.001	−0.545 * (−0.634–−0.441)	0.001	0.701 (0.625–0.764)	0.001	0.645 * (0.555–0.720)	0.001	0.524 (0.415–0.617)	0.001
Factor 1	0.454 * (0.341–0.554)	0.001	−0.351 * (−0.465–−0.225)	0.001	0.375 (0.252–0.485)	0.001	474 * (0.357–0.576)	0.001	0.419 (0.297–0.527)	0.001
Factor 2	0.633 * (0.545–0.707)	0.001	−0.606 * (−0.686–−0.512)	0.001	0.599 * (0.502–0.682)	0.001	0.536 * (0.428–0.629)	0.001	0.437 * (0.314–0.546)	0.001
Factor 3	0.523 * (0.419–0.614)	0.001	−0.380 * (−0.491–−0.257)	0.001	0.649 * (0.561–0.723)	0.001	0.443 * (0.323–0.550)	0.001	0.291 * (0.155–0.416)	0.001
Factor 4	0.532 * (0.429–0.622)	0.001	−0.457 * (−0.559–−0.342)	0.001	517 * (0.408–0.613)	0.001	0.457 * (0.338–0.562)	0.001	0.368 * (0.238–0.562)	0.001
Factor 5	0.409 * (0.292–0.514)	0.001	−0.294 * (−0.414–−0.164)	0.001	0.553 * (0.449–0.643)	0.001	0.438 * (0.317–0.545)	0.001	0.337 * (0.204–0.458)	0.001
Factor 6	0.299 * (0.173–0.416)	0.001	−0.262 * (−0.385–−0.130)	0.001	0.348 (0.223–0.461)	0.001	0.353 * (0.223–0.470)	0.001	0.256 * (0.117–0.385)	0.001
Factor 7	0.255 * (0.126–0.376)	0.001	−0.157 * (−0.287–−0.021)	0.020	0.370 * (0.243–0.484)	0.001	0.318 * (0.185–0.439)	0.001	0.311 * (0.176–0.435)	0.001

* Kolmogorov–Smirnov test.

## Data Availability

The data presented in this study are available on request from the corresponding author. The data are not publicly available due to ethical concerns.

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
