# Peer review of "Development and Psychometric Validation of the Breast Cancer Stigma Assessment Scale for Women with Breast Cancer and Its Survivors"

_healthcare, 2024, doi:10.3390/healthcare12040420_

Round 1
Reviewer 1 Report
Comments and Suggestions for Authors
Check the keywords based on the mesh.
Line 25, what’s ERI??
This doi: 10.3389/fonc.2022.921015. eCollection 2022. Can be useful to improve the introduction.
Line 71 -90 is confusing and not necessary.
Fig 1. Not needed.
In introduction, gap and purpose should be expressed in better sentence.
If possible, the introduction should be more concise.
Rewrite the method based on the standard checklist, the sequence of the contents is not appropriate.
An observational, descriptive, cross-sectional, and comparative study , this is confusing.
2.2 Instrument development The development of the instrument was based on the health-related stigma (HRS) theoretical model by Weiss [48] and was developed for cancer stigma in the previously 157 described conceptual framework by Fujisawa and Hagiwara [14]. It’s means, the questioner extracted from this?
an in-depth review: systematic ?
In parallel, a descriptive qualitative study… descriptive not needed.
participation of 15 women, why 15?
The pilot sample had 62 subjects, why 62?
2.3.3. Tools When the aim is to standardize the instrument, What was the purpose of these questionnaires?
What is the final questionnaire? In terms of the number of questions, grading, etc.
In the text, the reference number is above 80, But only 69 references have been reported.
I feel, the article is extracted from a general report with many objectives.
Author Response
Response to Review 1 Report (Round 1)
Article title: Development and psychometric validation of the Breast Cancer Stigma Assessment Scale for women with breast cancer and survivors.
First of all, I would like to thank the reviewers for their comments and suggestions. They have certainly contributed to improving the quality and readability of the manuscript and have clarified the way in which new ideas may be introduced. All of your suggestions have been incorporated into the manuscript. We hope that the modifications to the manuscript are suitable for publication
Reviewer 1
Reviewer (R): Check the keywords based on the mesh.
Authors (A): The keywords have been revised following your indication and two words have been modified. The keywords are as follows:
Line 38: “Keywords: Breast neoplasm; Cancer; Social Stigma; Patients; Cancer Survivors; Validation Study”
(R): Line 25, what’s ERI??
(A): Dear reviewer, we did not find the ERI reference in the document. Please let us know if, after the changes made to the manuscript, you still have doubts in this regard.
(R): This doi: 10.3389/fonc.2022.921015. eCollection 2022. Can be useful to improve the introduction.
(A): Thank you for this reference; following your recommendation we have added the reference:
Line 42-43: “Breast cancer, particulary, female breast cancer (FBC), is the most commonly diagnosed cancer in the world [1], and has the highest incidence and years lived with disability rate in Europe [2] , with an estimated 35,000 new cases in Spain by 2023 “
Allahqoli, L.; Mazidimoradi, A.; Momenimovahed, Z.; Rahmani, A.; Hakimi, S.; Tiznobaik, A.; Gharacheh, M.; Salehiniya, H.; Babaey, F.; Alkatout, I. The Global Incidence, Mortality, and Burden of Breast Cancer in 2019: Correlation With Smoking, Drinking, and Drug Use. Front Oncol 2022, 12, doi:10.3389/fonc.2022.921015.
(R): Line 71 -90 is confusing and not necessary.
(A): As suggested, we have deleted lines 71-90 in order to simplify the text.
(R): Fig 1. Not needed.
(A): We think it is necessary to keep Figure 1 as it gives a complete view of how stigma works and we believe that it will provide the reader with a conceptual map that may be useful in understanding the construct under study. We have added in line 79 a derivation to the concept map for better understanding of the text.
(R): In introduction, gap and purpose should be expressed in better sentence.
(A): In introduction, gap and purpose has been expressed in better sentence as follow:
Line 121-128: “The comorbidities associated with stigma justify the need to assess its role as a mediator or potential cause of decreased resilience, quality of life, amplification of morbidity and worsening prognosis. Stigma research will be useful in promoting and evaluating specific interventions that respond to the needs of women with breast cancer and survivors.
The aim of this study is to develop and validate a scale sensitive to the stigma experiences of Spanish women with breast cancer and survivors to explore its impact, incidence, duration, evolution and related factors”
(R): If possible, the introduction should be more concise.
(A): In line with your recommendations on this point and others, together with the recommendations of reviewer 3, we have modified the text to make it clearer and shorter.
(R): Rewrite the method based on the standard checklist, the sequence of the contents is not appropriate.
(A): When developing the methodology of the study, we reviewed several studies on the creation and validation of similar instruments, as well as the recommendations on good practices published by different authors (*some references considered). The structure and sequence of the proposed methodology were in line with the structure and steps required by the procedures and chronology of BCSAS development and validation. However, it is possible to improve this sequence, and following your suggestion, it has been reorganised as follows:
Line 133-135: “This study was divided into three phases: (1) Item development, (2) Scale development, (3) Scale evaluation following the recommendations for developing and validating scales [48–50].
Line 137: (Phase 1: Item Development)
Line 138-140: The development of the items was based on the health-related stigma (HRS) theoretical model by Weiss (Weiss et al., 2006) and was developed for cancer stigma in the previously described conceptual framework by Fujisawa and Hagiwara (Fujisawa & Hagiwara, 2015).
Line 191: 2.2.3. Delphi Study
Line 204: 2.3 Phase 2: Scale Development
Line 220: 2.4 Phase 3: Scale evaluation
Line 221: 2.4.1. Sample and setting
Line 229: 2.4.2. Data collection
Line 243: 2.4.3. Tools
Line 299: 2.4.3. Data analysis
Line 320: 2.5. Ethical consideration
* Bu, X., Li, S., Cheng, A. S. K., Ng, P. H. F., Xu, X., Xia, Y., & Liu, X. (2022). Breast Cancer Stigma Scale: A Reliable and Valid Stigma Measure for Patients With Breast Cancer. Frontiers in psychology, 13, 841280. https://doi.org/10.3389/fpsyg.2022.841280
*Morales-Asencio, J. M., Porcel-Gálvez, A. M., Oliveros-Valenzuela, R., Rodríguez-Gómez, S., Sánchez-Extremera, L., Serrano-López, F. A., Aranda-Gallardo, M., Canca-Sánchez, J. C., & Barrientos-Trigo, S. (2015). Design and validation of the INICIARE instrument, for the assessment of dependency level in acutely ill hospitalised patients. Journal of clinical nursing, 24(5-6), 761–777. https://doi.org/10.1111/jocn.12690
*Franke, M. F., Nelson, A. K., Muñoz, M., Cruz, J. S., Atwood, S., Lecca, L., & Shin, S. S. (2015). Validation of 2 Spanish-Language Scales to Assess HIV-Related Stigma in Communities. Journal of the International Association of Providers of AIDS Care, 14(6), 527–535. https://doi.org/10.1177/2325957414547738
*Cataldo, J. K., Slaughter, R., Jahan, T. M., Pongquan, V. L., & Hwang, W. J. (2011). Measuring stigma in people with lung cancer: psychometric testing of the cataldo lung cancer stigma scale. Oncology nursing forum, 38(1), E46–E54. https://doi.org/10.1188/11.ONF.E46-E54
*Clark, L. A., & Watson, D. (2019). Constructing validity: New developments in creating objective measuring instruments. Psychological assessment, 31(12), 1412–1427. https://doi.org/10.1037/pas0000626
*Boateng, G. O., Neilands, T. B., Frongillo, E. A., Melgar-Quiñonez, H. R., & Young, S. L. (2018). Best Practices for Developing and Validating Scales for Health, Social, and Behavioral Research: A Primer. Frontiers in public health, 6, 149. https://doi.org/10.3389/fpubh.2018.00149
*Dima A. L. (2018). Scale validation in applied health research: tutorial for a 6-step R-based psychometrics protocol. Health psychology and behavioral medicine, 6(1), 136–161. https://doi.org/10.1080/21642850.2018.1472602
(R) An observational, descriptive, cross-sectional, and comparative study, this is confusing.
(A) It has been amended as follows:
Line 132-133: “A descriptive, cross-sectional study was carried out for the development and validation of an instrument.”
(R): 2.2 Instrument development The development of the instrument was based on the health-related stigma (HRS) theoretical model by Weiss [48] and was developed for cancer stigma in the previously 157 described conceptual framework by Fujisawa and Hagiwara [14]. It’s means, the questioner extracted from this?
(A): No, it isn't. The Weiss theoretical model developed for cancer by Fujisawa and Hagiwara was used to structure and delimit the initial selection of items drawn from the above scales (HIV and lung cancer) and from the results of the qualitative research. The scale items were developed and selected by inductive and deductive methods, which is the way recommended by the literature. (Boateng et al., 2018)
(R): an in-depth review: systematic?
(A): The review you indicate was a comprehensive and systematised review, but we cannot consider it a systematic review as it does not meet all the criteria to be considered as such according to the PRISMA guidelines.
PRISMA guidelines reference: Page, M. J., McKenzie, J. E., Bossuyt, P. M., Boutron, I., Hoffmann, T. C., Mulrow, C. D., Shamseer, L., Tetzlaff, J. M., Akl, E. A., Brennan, S. E., Chou, R., Glanville, J., Grimshaw, J. M., Hróbjartsson, A., Lalu, M. M., Li, T., Loder, E. W., Mayo-Wilson, E., McDonald, S., ... Moher, D. (2021). The PRISMA 2020 statement: An updated guideline for re-porting systematic reviews. BMJ (Clinical Research Ed.), 372. https://doi.org/10.1136/BMJ.N71Poon,
(R): In parallel, a descriptive qualitative study… descriptive not needed.
(A): Following your recommendation, the word descriptive has been deleted.
Line 148: “In parallel, a qualitative study was conducted with the participation of 15 women with breast cancer or survivors over 18 years of age...”
(R): participation of 15 women, why 15?
(A): Thank you for your question, the sample size was reached when saturation of the qualitative data was obtained. A clarification has been added in the text.
Line 152: “The sample size was reached at the saturation of the data.”
(R): The pilot sample had 62 subjects, why 62?
(A): The literature states that the minimum sample size for conducting an exploratory factor analysis is 50-60 cases (Comrey & Lee, 1992; de Winter et al., 2009; Sapnas & Zeller, 2002). In our case, we sought to obtain 60, but due to circumstances of the sample collection, we finally obtained 62. This cut-off was made to ensure that the preliminary results could show us how the instrument worked as a pilot, in order to check the psychometric characteristics of the instrument and to be able to detect any aspect to be modified before collecting the sample of the scale.
Comrey, A. L., & Lee, H. B. (1992). A first course in factor analysis, 2nd ed. In A first course in factor analysis, 2nd ed. Lawrence Erlbaum Associates, Inc.
De Winter, J. C. F., Dodou, D., & Wieringa, P. A. (2009). Exploratory factor analysis with small sample sizes. Multivariate Behavioral Research, 44(2), 147–181. https://doi.org/10.1080/00273170902794206
Sapnas, K. G., & Zeller, R. A. (2002). Minimizing sample size when using exploratory factor analysis for measurement. Journal of Nursing Measurement, 10(2), 135–154. https://doi.org/10.1891/JNUM.10.2.135.52552
(R): 2.3.3. Tools When the aim is to standardize the instrument, What was the purpose of these questionnaires?
(A): When validating a scale that assesses a construct for which there is no already validated instrument available that assesses the same construct (gold standard), it is necessary to check that the instrument correlates with other instruments that relate to the construct you are assessing. Since breast cancer stigma is related to anxiety, depression, quality of life, centrality of the event, etc. and in our population, we have scales that assess these aspects, these instruments were selected to assess convergent validity with our instrument (BCSAS).
(R): What is the final questionnaire? In terms of the number of questions, grading, etc.
(A): The final scale is described in the tools section of the methodology, to which we have added a clarification in the text to make it easier to understand. In the next analysis, with a larger sample, we plan to estimate cut-offs at which stigma poses a risk to the quality of life and health of women with breast cancer and survivors.
Line 249-252: “Breast Cancer Stigma Assessment Scale (BCSAS): this scale has 28 items, which consisted of rating statements of stigma experiences with a 5-point Likert scale (1 = strongly disagree and 5 = strongly agree). It was scored between a minimum of 28 and a maximum of 140; the higher the score, the greater the degree of stigma”.
(R): In the text, the reference number is above 80, But only 69 references have been reported.
(A): A malfunction in the Mendeley configuration caused this inconsistency. It has already been resolved in the text and in the bibliography, which has 76 references cited.
(R): I feel, the article is extracted from a general report with many objectives.
(A): The aim of this article is to develop and validate an instrument. All the steps that have been followed have been carried out with this sole objective in mind. This is the first article of the first author's doctoral thesis. This article has developed the first objective which, as stated, was to develop and validate an instrument to measure and assess the stigma of Spanish women with breast cancer and survivors. The second objective, to characterise stigma using the BCSAS, is the subject of another article that is currently being prepared.

Reviewer 2 Report
Comments and Suggestions for Authors
Dear authors
I would like to thank you for giving me the opportunity to review the manuscript entitled “Development and psychometric validation of the Breast Cancer Stigma Assessment Scale for women with breast cancer and survivors”. The aim of this study was to develop and validate a breast cancer stigma scale in Spanish context. This manuscript is well-designed and written and generates a valuable tool.
- Line 313: Data analysis
- In the validation phase, are 231 participants enough for an instrument with 28 items?
Author Response
Response to Review 2 Report (Round 1)
Reviewer (R): I would like to thank you for giving me the opportunity to review the manuscript entitled “Development and psychometric validation of the Breast Cancer Stigma Assessment Scale for women with breast cancer and survivors”. The aim of this study was to develop and validate a breast cancer stigma scale in Spanish context. This manuscript is well-designed and written and generates a valuable tool.
Authors (A): Thank you very much for your comments. The authors believe that they have helped to improve the quality of the manuscript.
(R): Line 313: Data analysis
(A): Thank you for catching that typo. It has been corrected in the text
(R): In the validation phase, are 231 participants enough for an instrument with 28 items?
(A): On the one hand, there is no consensus on the minimum sample size for conducting an exploratory factor analysis. Several authors indicate that this should be a minimum of 200 (Boateng et al., 2018; Boomsma, 1982; Cattell, 1978; de Winter et al., 2009; Sapnas & Zeller, 2002); in our case, we obtained a sample of 231, which we consider sufficient for carrying out the analysis. Although other authors indicate that this sample size may be suboptimal in some cases (Boomsma, 1982). On the other hand, our scale has a larger sample size than other validated breast cancer stigma scales (Bu et al., 2022; Dewan et al., 2020). The scale adapted in China had a sample size of 200 (Dewan et al., 2020), while the scale in Arab patients was validated in 59 patients (Dewan et al., 2020). These sample sizes are due to the difficulty of accessing the study population, which limits the collection of a larger sample.
Boomsma, A. (1982). The robustness of LISREL against small sample sizes in factor analysis models. Part I, 1. https://www.econbiz.de/Record/the-robustness-of-lisrel-against-small-sample-sizes-in-factor-analysis-models-boomsma/10001936630
Boateng, G. O., Neilands, T. B., Frongillo, E. A., Melgar-Quiñonez, H. R., & Young, S. L. (2018). Best Practices for Developing and Validating Scales for Health, Social, and Behavioral Research: A Primer. Frontiers in public health, 6, 149. https://doi.org/10.3389/fpubh.2018.00149
Bu, X., Li, S., Cheng, A. S. K., Ng, P. H. F., Xu, X., Xia, Y., & Liu, X. (2022). Breast Cancer Stigma Scale: A Reliable and Valid Stigma Measure for Patients With Breast Cancer. Frontiers in Psychology, 13. https://doi.org/10.3389/fpsyg.2022.841280
Cattell, R. B. (1978). The Scientific Use of Factor Analysis in Behavioral and Life Sciences. The Scientific Use of Factor Analysis in Behavioral and Life Sciences. https://doi.org/10.1007/978-1-4684-2262-7
Comrey, A. L., & Lee, H. B. (1992). A first course in factor analysis, 2nd ed. In A first course in factor analysis, 2nd ed. Lawrence Erlbaum Associates, Inc.
de Winter, J. C. F., Dodou, D., & Wieringa, P. A. (2009). Exploratory factor analysis with small sample sizes. Multivariate Behavioral Research, 44(2), 147–181. https://doi.org/10.1080/00273170902794206
Dewan, M., Hassouneh, D., Song, M. K., & Lyons, K. (2020). Development of the Breast Cancer Stigma Scale for Arab Patients. Asia-Pacific Journal of Oncology Nursing, 7(3), 295–300. https://doi.org/10.4103/apjon.apjon_14_20
Sapnas, K. G., & Zeller, R. A. (2002). Minimizing sample size when using exploratory factor analysis for measurement. Journal of Nursing Measurement, 10(2), 135–154. https://doi.org/10.1891/JNUM.10.2.135.52552
Reviewer 3 Report
Comments and Suggestions for Authors
I am very grateful to the authors for this excellent contribution to the journal and also their work towards improving the lives of women with and surviving breast cancer. The paper makes it clear that the authors have created a comprehensive and culturally appropriate scale to measure the stigma associated with breast cancer that is well validated. I highly commend the authors on their important work.
I have some suggestions about how the manuscript can be improved before publication, most (if not all) are minor points.
1. Is it possible to include the completed BCSAS instrument and/or an English translation with the published manuscript as an Appendix?
2. Tables: I would change the font of the tables to be consistent with the rest of the paper.
3. Tables 2 and 3: Is it possible to format these tables better? I think it will improve readability, especially if they could be fit in one page.
4. Conclusion: This is the weakest part of the paper. It is just one sentence. I think the authors should extend this section, referring back the introduction, where the authors excellently describe the study aims and the importance of studying and recognizing stigma.
5. Lines 538, Acknowledgements section needs to be completed.
6. There are some problems with the English. Although the level of the language is quite high, there are some phrases and particularly sentence structure/flow issues that stand out a bit to a native-speaker. The authors should consider having a native-speaker, preferably an academic with expertise in the field, edit the paper. Among the issues I noticed were some very long and meandering sentences; for example: Lines 118 -124, 130 – 135,162 – 166, 410-414 These sentences, and others, should be broken into smaller separate sentences for clarity and improved flow.
7. Line 134: There is a grammatical issue on line 134.
8. Lines 145 and 500: Change the abbreviation “BC” to “breast cancer” written in full
9. Lines 227, 350, 534: Change “subjects” to “participants”
10. Line 228: Write abbreviations in full at first mention: “Kaiser–Meyer–Olkin (KMO) value, there after just use the abbreviation
11. Lines 259 332: Add the manufacturer’s details (version, company name, country) for LimeSurvey software and SPSS.
12. In the discussion section, use square brackets [##] for references, some places you used curved brackets (##).
13. Section 2.3.3: Remove the hyphens from before each listed tool. No bullet points are needed here.
14. Line 313: Change to English: “análisis”
15. Line 338: Space needed after “information.”
16. Line 484: Closing parenthesis needed after “(95%, CI= 0.415-0.617”
Comments on the Quality of English LanguageI will repeat my 6th comment from above: There are some problems with the English. Although the level of the language is quite high, there are some phrases and particularly sentence structure/flow issues that stand out a bit to a native-speaker. The authors should consider having a native-speaker, preferably an academic with expertise in the field, edit the paper. Among the issues I noticed were some very long and meandering sentences; for example: Lines 118 -124, 130 – 135,162 – 166, 410-414 These sentences, and others, should be broken into smaller separate sentences for clarity and improved flow.
Author Response
Response to Review 3 Report (Round 1)
Reviewer (R): I am very grateful to the authors for this excellent contribution to the journal and also their work towards improving the lives of women with and surviving breast cancer. The paper makes it clear that the authors have created a comprehensive and culturally appropriate scale to measure the stigma associated with breast cancer that is well validated. I highly commend the authors on their important work.
I have some suggestions about how the manuscript can be improved before publication, most (if not all) are minor points.
Authors (A): Thank you very much for your words. The authors hope that you will find the following responses to your suggestions satisfactory.
(R): Is it possible to include the completed BCSAS instrument and/or an English translation with the published manuscript as an Appendix?
(A): In accordance with your suggestion, we have added an appendix with BCSAS in both languages, Spanish as the language of validation and English to facilitate its diffusion.
(R): Tables: I would change the font of the tables to be consistent with the rest of the paper.
(A): Following your recommendation, the font of all tables has been modified.
(R): Tables 2 and 3: Is it possible to format these tables better? I think it will improve readability, especially if they could be fit in one page.
(A): The structure of table 2 and 3 has been modified in relation to your suggestions.
(R): Conclusion: This is the weakest part of the paper. It is just one sentence. I think the authors should extend this section, referring back the introduction, where the authors excellently describe the study aims and the importance of studying and recognizing stigma.
(A):
(A): We have modified the conclusion at your suggestion as follows:
Line 508-512: “It is imperative that health professionals incorporate stigma assessment into their interventions. The associated comorbidities and the additional suffering involved for women with breast cancer and survivors are evidence of this. This will enable individualised responses and the development of health policies for breast cancer health promotion and prevention”.
(R): Lines 538, Acknowledgements section needs to be completed.
(A):
A: Thank you for this indication. We have completed the acknowledgements.
Line 530-536: “This study includes results from the doctoral thesis of the first author. Our thanks go to the Research Support Unit of the Hospital Virgen de las Nieves and the Hospital Vall de Hebrón, to the patient associations that have collaborated, especially AMAMA. Also to the experts who advised on the selection of items and supported this research. Our thanks to the people who have collaborated and supported this research, especially to all the women who have participated and made this research possible, for their commitment and truthfulness with the aim of helping future patients and survivors”
(R): There are some problems with the English. Although the level of the language is quite high, there are some phrases and particularly sentence structure/flow issues that stand out a bit to a native-speaker. The authors should consider having a native-speaker, preferably an academic with expertise in the field, edit the paper. Among the issues I noticed were some very long and meandering sentences; for example: Lines 118 -124, 130 – 135,162 – 166, 410-414 These sentences, and others, should be broken into smaller separate sentences for clarity and improved flow.
(A): Thanks to your feedback we have proceeded to simplify these sentences, as you suggest it will improve the reading flow.
Line 99-108: “Likewise, some reviews and meta-analyses in recent years have corroborated the strong relationship of stigma in women with breast cancer with the loss of body image, anxiety, resignation, depression, guilt, ambivalence about emotional expression, and social restriction, as well as with delayed help-seeking behavior [36,37]. It is showing that there are numerous sociodemographic variables, the disease itself and its treatments, and psychosocial variables that are related to stigma, which causes suffering and conditions the disease experience and recovery of these women.”
Line 11-120: “Based on the latter, the Cataldo Lung Cancer Stigma Scale (CLCSS) was developed in 2011 [14], and recently, instruments have been developed to specifically measure stigma in Iranian and Chinese women with breast cancer (Dewan et al. 2020 and Bu et al. 2022) [34,35]. However, these tools are designed for patients in the active phase. This kind of patient is focus on the consequences of cosmetic changes. These scales not include other relevant items for the assessment of stigma or its impact on survivors. n the other hand, we have not found any instrument developed in the Spanish population that assesses breast cancer stigma. Neither does it include all relevant aspects of stigma apart from the aesthetic items relevant to the assessment of stigma or its impact on survivors.
Line 395-402: “To our knowledge, two recently published scales have been developed to specifically assess breast cancer stigma. The Breast Cancer Stigma Scale for Arab patients (BCSS-A), published in 2020 [34]. It is a questionnaire without factorial structure composed of 12 items. This scale was developed with a sample of 59 Muslim women undergoing active cancer treatment, almost all of whom were married. The BCSS-A showed correlation only with depressive symptomatology. This scale has important methodological and conceptual limitations that prevent us from generalizing its findings and, therefore, from comparing results.”
(R): Line 134: There is a grammatical issue on line 134.
(A): Thank you for your comment, we have modified the sentence to correct the typo.
Line 114-115: “However, these tools are designed for patients in the active phase. This kind of patient is focus on the consequences of cosmetic changes.”
(R): Lines 145 and 500: Change the abbreviation “BC” to “breast cancer” written in full
(A): The acronyms have been modified according to your recommendations.
Line 126-128: “The aim of this study is to develop and validate a scale sensitive to the stigma experiences of Spanish women with breast cancer and survivors to explore its impact, incidence, duration, evolution and related factors.”
Line 485-487: “The availability of this instrument could facilitate the incorporation of stigma into the health policy agenda and serve to promote and evaluate specific interventions on stigma in breast cancer.”
(R): Lines 227, 350, 534: Change “subjects” to “participants”
(A): The text has been modified according to your suggestions
Line 212-215: “The pilot sample had 62 participants, showing an α = 0.883 (95%, CI =0.836-0.921) with a Kaiser–Meyer–Olkin (KMO) value = 0.746 and a Bartlett's sphericity test value of χ2 = 758.981 (p = 0.001). This indicated that the BCSAS had a good internal consistency and the exploratory factor analysis was relevant”
Line 335-336: “Regarding the feasibility, the mean response time assessed in 68 participants was 7.10 min (SD = 5.39).”
Line 526-527: “Informed consent was obtained from all participants involved in the study”
(R): Line 228: Write abbreviations in full at first mention: “Kaiser–Meyer–Olkin (KMO) value, there after just use the abbreviation
(A): The text has been modified according to your comment.
Line 212-215: “The pilot sample had 62 participants, showing an α = 0.883 (95%, CI =0.836-0.921) with a Kaiser–Meyer–Olkin (KMO) value = 0.746 and a Bartlett's sphericity test value of χ2 = 758.981 (p = 0.001). This indicated that the BCSAS had a good internal consistency and the exploratory factor analysis was relevant”
(R): Lines 259 332: Add the manufacturer’s details (version, company name, country) for LimeSurvey software and SPSS.
(A): We have proceeded to cite the computer programs in the bibliography. The type of bibliographic format we use does not present some of the data you request, which are detailed below:
Limesurvey GmbH. / LimeSurvey: An Open Source survey tool /LimeSurvey GmbH, Hamburg, Germany. URL http://www.limesurvey.or
LimeSurvey Community Edition
Versión 5.6.31+230718
IBM Corp. IBM SPSS Statistics for Windows 2017. V28.
(R): In the discussion section, use square brackets [##] for references, some places you used curved brackets (##).
(A): We have fixed that typo.
(R): Section 2.3.3: Remove the hyphens from before each listed tool. No bullet points are needed here.
(A): Thank you for this recommendation; we have removed the hyphens.
(R): Line 313: Change to English: “análisis”
(A): Thank you for catching that typo. It has been corrected in the text
(R): Line 338: Space needed after “information.”
(A): The typo has been modified
(R): Line 484: Closing parenthesis needed after “(95%, CI= 0.415-0.617”
(A): Thanks, the typo has been modified
(R): Comments on the Quality of English Language
I will repeat my 6th comment from above: There are some problems with the English. Although the level of the language is quite high, there are some phrases and particularly sentence structure/flow issues that stand out a bit to a native-speaker. The authors should consider having a native-speaker, preferably an academic with expertise in the field, edit the paper. Among the issues I noticed were some very long and meandering sentences; for example: Lines 118 -124, 130 – 135,162 – 166, 410-414 These sentences, and others, should be broken into smaller separate sentences for clarity and improved flow.
(A): Thank you for this input. We have simplified the suggested sentences to improve reading flow and comprehension.

Round 2
Reviewer 1 Report
Comments and Suggestions for Authors
-